# Eosinophilic Liver Abscess Mimicking Intrahepatic Cholangiocarcinoma on 18F-FDG PET-CT: A Case Report

**DOI:** 10.3390/diagnostics15233057

**Published:** 2025-11-29

**Authors:** Hongzhen Wei, Yifan Ma, Yizhuo Shi, Erming Cai, Shiran Sun, Chidan Wan

**Affiliations:** Department of Hepatobiliary Surgery, Union Hospital, Tongji Medical College, Huazhong University of Science and Technology, Wuhan 430022, China; d202582583@hust.edu.cn (H.W.); m202376263@hust.edu.cn (Y.M.); m202476308@hust.edu.cn (Y.S.); m202376248@hust.edu.cn (E.C.)

**Keywords:** eosinophilic liver abscess, intrahepatic cholangiocarcinoma, PET-CT, eosinophilia, diagnostic pitfall, case report

## Abstract

Background and Clinical Significance: Eosinophilic liver abscess (ELA) is a rare benign condition that can mimic malignant liver tumors on imaging studies. The diagnostic challenge is further compounded when 18F-FDG PET-CT demonstrates high metabolic activity. Case Presentation: A 39-year-old man presented with abdominal pain, jaundice, and marked peripheral eosinophilia (34%). Imaging revealed a single liver mass with intense FDG uptake (SUVmax 10.8), highly suspicious for intrahepatic cholangiocarcinoma (ICC). Parasitic serologies were negative. Surgical resection revealed eosinophilic infiltration without malignancy. The patient remained disease-free at 1-year follow-up. Conclusions: ELA can closely mimic ICC on PET-CT, with high FDG uptake representing a significant diagnostic pitfall. This case underscores the importance of considering ELA in the differential diagnosis of hypermetabolic liver masses in patients with peripheral eosinophilia and highlights that surgical resection remains a reasonable approach when malignancy cannot be excluded.

## 1. Introduction

Eosinophilic liver abscess (ELA) is a rare benign condition characterized by localized eosinophilic infiltration in the liver parenchyma, often with central necrosis [1]. The etiology is diverse, including parasitic infections, drug hypersensitivity, malignancy-associated eosinophilia, and idiopathic causes [2]. A large retrospective study from Korea reported an incidence of 0.68% among patients undergoing abdominal CT, suggesting ELA may be more common than previously recognized [3].

The clinical presentation ranges from asymptomatic incidental findings to symptomatic cases with abdominal pain, fever, and weight loss. Laboratory findings typically include peripheral eosinophilia, though the degree varies considerably [4].

Understanding the clinical and imaging features of ELA is crucial to avoid unnecessary aggressive treatments while ensuring appropriate management when malignancy cannot be excluded. We report a case of idiopathic eosinophilic liver abscess in a 59-year-old man that closely mimicked ICC on both conventional imaging and 18F-FDG PET-CT, with a notably high SUVmax of 10.8. This case highlights the diagnostic pitfall of PET-CT in eosinophilic liver lesions and the role of surgical resection in selected patients.

## 2. Case Presentation

### 2.1. Clinical History

Referred from a secondary hospital, a 39-year-old Chinese man presented to our hospital with a 2-month history of intermittent right upper quadrant abdominal pain, progressive jaundice, and weight loss of approximately 5 kg. The abdominal pain was dull and aching, without radiation. Outside CT suggested mild left intrahepatic ductal dilatation with unclear hilar anatomy and no gallbladder/CBD stones. He also reported dark-colored urine and pale stools. There was no history of fever, pruritus, or recent travel to endemic areas.

The patient had a history of hypertension, well-controlled with medication. He denied allergies, parasitic infections, raw food consumption, or contact with livestock. He was a non-smoker and consumed alcohol occasionally. There was no family history of malignancy or liver disease.

On physical examination, the patient appeared jaundiced with scleral icterus. Vital signs were stable. Abdominal examination revealed mild tenderness in the right upper quadrant. The liver edge was palpable 2 cm below the right costal margin, with a smooth surface and firm consistency. No splenomegaly or ascites were detected.

### 2.2. Laboratory Investigations

Laboratory investigations revealed marked peripheral eosinophilia and cholestatic liver dysfunction (Table 1).

The markedly elevated eosinophil percentage (34%) and cholestatic pattern, combined with significantly elevated CA125 (199.7 U/mL), raised concerns for either parasitic infection or malignancy.

### 2.3. Imaging Studies

CT revealed patchy low-density lesions in the left hepatic lobe and hilar region, with mild intrahepatic bile duct dilatation and local bile duct wall thickening. The largest lesion was about 14 mm in diameter. Multiple enlarged lymph nodes were seen in the hepatic hilum and hepatoduodenal ligament areas. Malignancy, such as cholangiocarcinoma, was suspected (Figure 1).

MRI showed a mass-like lesion in the left hepatic hilum, about 30 × 22 mm, with ill-defined borders and marked enhancement after contrast. Dilatation of intrahepatic bile ducts and multiple enlarged lymph nodes in the hepatic hilum were also observed (Figure 2).

The left hepatic lobe near the hilum shows a mass-like lesion with slightly high signal intensity on T2WI and mild perilesional edema.

PET-CT showed a mass in the left hepatic hilum region, with intense FDG uptake (SUVmax 10.8). The intrahepatic bile ducts were dilated, especially on the left side. Multiple enlarged lymph nodes in the hilar region showed increased FDG uptake (SUVmax 3.9–5.6, up to 1.5 × 0.9 cm in size). No abnormal FDG uptake was found in other organs (Figure 3).

The imaging findings strongly suggested intrahepatic cholangiocarcinoma with regional lymph node involvement.

### 2.4. Multidisciplinary Discussion and Surgical Intervention

The case was discussed at a multidisciplinary tumor board. Given the imaging features highly suggestive of ICC, elevated CA125, high SUVmax on PET-CT, and inability to exclude malignancy despite marked eosinophilia, surgical resection was recommended for both diagnostic and therapeutic purposes. To manage the jaundice and bile duct dilatation preoperatively, the patient had undergone percutaneous transhepatic cholangiodrainage (PTCD) for biliary decompression; no additional stenting was performed, as the surgical plan included resection and biliary reconstruction to directly alleviate the compressive obstruction.

The patient underwent laparoscopic exploration, which was subsequently converted to open left hemihepatectomy (segments II–IV), along with caudate lobe (segment I) resection, cholecystectomy, hepatic duct plasty, hepaticojejunostomy, and regional lymph node dissection. Intraoperatively, a firm, well-circumscribed mass was identified at the confluence of the left and right hepatic ducts, closely adherent to the bile duct and portal vein branches, with invasion of the hepatic duct confluence but sparing the right secondary branches. The mass and involved structures were completely resected, relieving biliary compression. Multiple enlarged lymph nodes in the hepatoduodenal ligament were excised. No vascular invasion was noted, and biliary drainage was restored via hepaticojejunostomy.

### 2.5. Pathological Findings

Gross examination revealed a well-circumscribed mass measuring 4.5 × 2.0 × 1.0 cm (with other nodules measuring up to 8 × 4 cm). Microscopic examination showed dense inflammatory infiltration predominantly composed of eosinophils and histiocytes, with areas of necrosis. No parasitic organisms or malignant cells were identified. Immunohistochemical staining showed CD30 (partial+), CD68 (partial+), S-100 (scattered+), PCK (−), CD34 (−), CD21 (−), ALK (−), CD1a (−), Langerin (−), and Ki-67 (about 10%) (Figure 4).

Multiple regional lymph nodes from the hepatic hilum and gallbladder neck were examined. Histological examination revealed reactive hyperplasia without evidence of malignancy or parasitic infection. The lymph node status was negative for metastatic disease. The lymph node status was negative.

The findings ruled out epithelial malignancy. The final pathological diagnosis was inflammatory myofibroblastic proliferation with eosinophilic and histiocytic infiltration, consistent with an eosinophilic liver abscess with reactive lymphadenopathy.

### 2.6. Postoperative Course and Follow-Up

The patient had an uneventful postoperative recovery. Jaundice gradually resolved, and liver function tests mostly normalized within 2 weeks, with some cholestatic markers (ALP, GGT) remaining mildly elevated. Eosinophil count decreased to 12% by postoperative day 7 and normalized to 5.7% by 1 month. The patient was discharged on postoperative day 8. No specific anti-parasitic or anti-inflammatory treatment was administered.

At 1-year follow-up, the patient remained asymptomatic, with normal liver function tests and eosinophil counts. Repeat CT imaging showed no evidence of recurrence or new lesions.

## 3. Discussion

### 3.1. Etiology and Clinical Spectrum of Eosinophilic Liver Lesions

Eosinophilic liver abscess is a rare condition with diverse etiologies. To better understand the clinical spectrum, we reviewed recent literature and identified several representative cases (Table 2).

Among parasitic causes, *Toxocara canis* and *Fasciola hepatica* are most commonly reported [5,6,9]. In the Korean series, 48.2% of cases were attributed to parasitic infections, while the remaining had unidentified etiologies or were associated with malignancies [3]. Malignancy-associated ELA, as exemplified by Shigematsu et al. [8], occurs when primary tumors produce eosinophilic chemotactic factors transported to the liver via portal blood flow.

Our case is notable for the absence of identifiable parasitic infection despite extensive serological testing and stool ova examination, and no evidence of malignancy on imaging and follow-up, suggesting an idiopathic form of ELA. The eosinophil percentage of 34% is among the highest reported and comparable to the Fasciola hepatica case (35.3%) [5].

### 3.2. The Diagnostic Pitfall of PET-CT in Eosinophilic Liver Lesions

One of the most challenging aspects of our case is the high FDG uptake (SUVmax 10.8) on PET-CT, a value typically associated with malignancy. Li et al. [7] recently reported a case of eosinophilic liver infiltration with high FDG uptake, where 18F-FAPI PET/CT showed even higher tracer uptake than FDG, both mimicking malignancy. This finding suggests that both FDG and FAPI PET/CT can yield false-positive results in eosinophilic liver lesions.

The high FDG uptake can be explained by the intense inflammatory activity and high metabolic rate of eosinophils and other inflammatory cells [10]. Eosinophils have high glucose metabolism and can accumulate FDG, leading to false-positive PET-CT results.

Our case underscores that high FDG uptake (even SUVmax > 10) does not automatically indicate malignancy in patients with peripheral eosinophilia. Clinicians should be aware of this diagnostic pitfall and consider ELA in the differential diagnosis of hypermetabolic liver masses.

### 3.3. Unique Features of Our Case

Compared with previously reported cases of ELA, our case presents several notable and distinctive features. First, the patient exhibited significant cholestatic liver dysfunction (total bilirubin 51.3 μmol/L), which differs from most reported cases in which liver function is normal [5,6,8]. The likely cause is compression of the bile duct by the mass and surrounding inflammation. In addition, the lesion in our case was solitary, whereas many cases in the literature involve multiple lesions [3,6,7]; solitary lesions are more likely to be misdiagnosed as malignant on imaging. Finally, and perhaps most challenging, PET-CT in this case demonstrated a SUVmax as high as 10.8, which is among the highest values reported for ELA and substantially increased the diagnostic difficulty.

### 3.4. Differential Diagnosis and Diagnostic Approach

For hypermetabolic liver masses accompanied by peripheral eosinophilia, the differential diagnosis is broad. In this case, considering the features of a solitary mass, cholestasis, elevated CA125, and high SUVmax, intrahepatic cholangiocarcinoma (ICC) was the primary diagnosis; however, ICC generally does not present with such pronounced eosinophilia. Another key differential diagnosis is parasitic liver abscess, as Toxocara, Fasciola hepatica, Echinococcus, and Schistosoma can all cause eosinophilic liver lesions, making serological testing essential. Additionally, malignancy-associated eosinophilic abscesses should be considered, since some solid tumors—especially gastrointestinal malignancies—can induce eosinophilia and hepatic eosinophilic infiltration [8]. Rare foreign body-related hepatic lesions should also be considered, as migrated ingested objects can perforate the gastrointestinal tract and reach the liver, producing localized inflammation/abscesses that mimic hepatic tumors on imaging [11]. Lastly, idiopathic hypereosinophilic syndrome, characterized by persistent eosinophilia with end-organ damage, remains a diagnosis of exclusion [12].

In our patient, after extensive investigations—including parasitic serology, whole-body imaging, and comprehensive review of medication history—no definitive etiology could be identified, leading to a final diagnosis of idiopathic ELA.

### 3.5. Treatment Strategies and the Role of Surgical Resection

Treatment of ELA depends on the underlying etiology. For parasitic infections, specific anti-parasitic therapy is highly effective, with most cases showing complete resolution within weeks to months (Table 2). For malignancy-associated ELA, treatment of the primary tumor often leads to spontaneous resolution of hepatic lesions.

However, in our case, the solitary nature of the lesion, inability to exclude malignancy, and presence of symptoms made surgical resection the most appropriate approach. This decision aligns with current clinical practice for suspected malignant liver tumors [13].

The outcome of this case underscores that surgical resection can serve as both an effective diagnostic and therapeutic strategy for selected patients with ELA. Specifically, surgical intervention becomes a reasonable option when malignancy cannot be definitively excluded based on imaging and clinical features. This approach is particularly justified in situations where the lesion is solitary and anatomically suitable for resection, the patient presents with significant symptoms, and percutaneous biopsy is either technically unfeasible or yields inconclusive results. In such complex clinical scenarios, surgery not only provides a definitive pathological diagnosis but also achieves complete removal of the lesion, thereby fulfilling both diagnostic and therapeutic objectives.

Importantly, surgical resection in our case not only provided a definitive diagnosis but also achieved complete cure, with no recurrence at 1-year follow-up. This outcome is comparable to or better than medically managed cases (Table 2), suggesting that surgery is a valid treatment option for selected patients with ELA.

## 4. Conclusions

We report a rare case of idiopathic eosinophilic liver abscess (ELA) that closely mimicked intrahepatic cholangiocarcinoma (ICC) with a high SUVmax of 10.8 on 18F-FDG PET-CT. This case highlights a significant diagnostic pitfall, as the intense metabolic activity of eosinophilic inflammation can lead to high FDG uptake, falsely suggesting malignancy. Therefore, ELA should be considered in the differential diagnosis of hypermetabolic liver masses in patients with peripheral eosinophilia. When malignancy cannot be excluded, surgical resection remains a reasonable approach, serving both diagnostic and therapeutic purposes and leading to excellent outcomes. Despite its aggressive appearance on imaging, ELA has a benign prognosis with appropriate management, underscoring the importance of a high index of suspicion and pathological confirmation for accurate diagnosis and treatment.

## Figures and Tables

**Figure 1 diagnostics-15-03057-f001:**
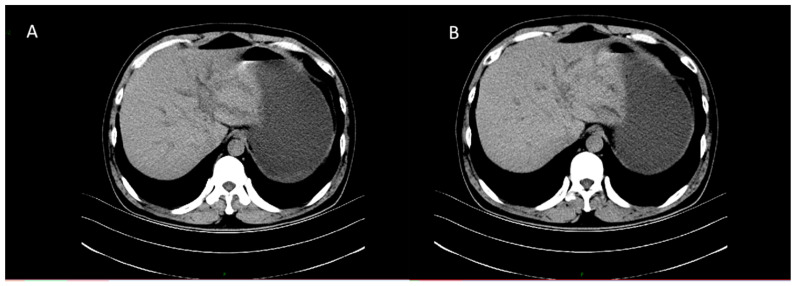
Axial contrast-enhanced CT images of the liver. (**A**) Plain scan shows a patchy low-density lesion in the left lobe of the liver near the hepatic hilum. (**B**) Arterial phase shows mild heterogeneous enhancement of the lesion.

**Figure 2 diagnostics-15-03057-f002:**
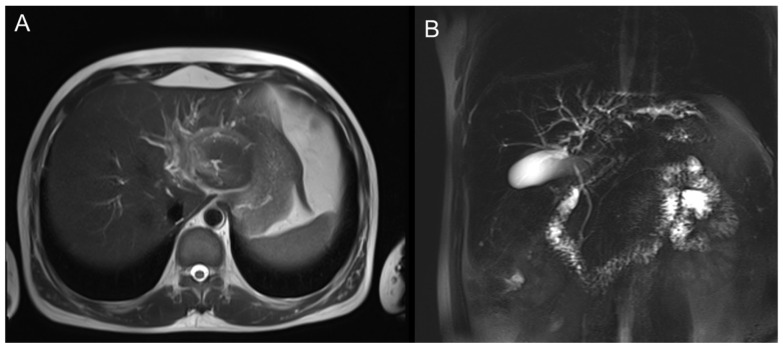
Axial T2-weighted MR images of the liver. (**A**) Axial view demonstrating a well-defined hilar lesion with dilated intrahepatic bile ducts. (**B**) Coronal view showing the hepatic hilar mass with intrahepatic biliary dilatation and gallbladder distension.

**Figure 3 diagnostics-15-03057-f003:**
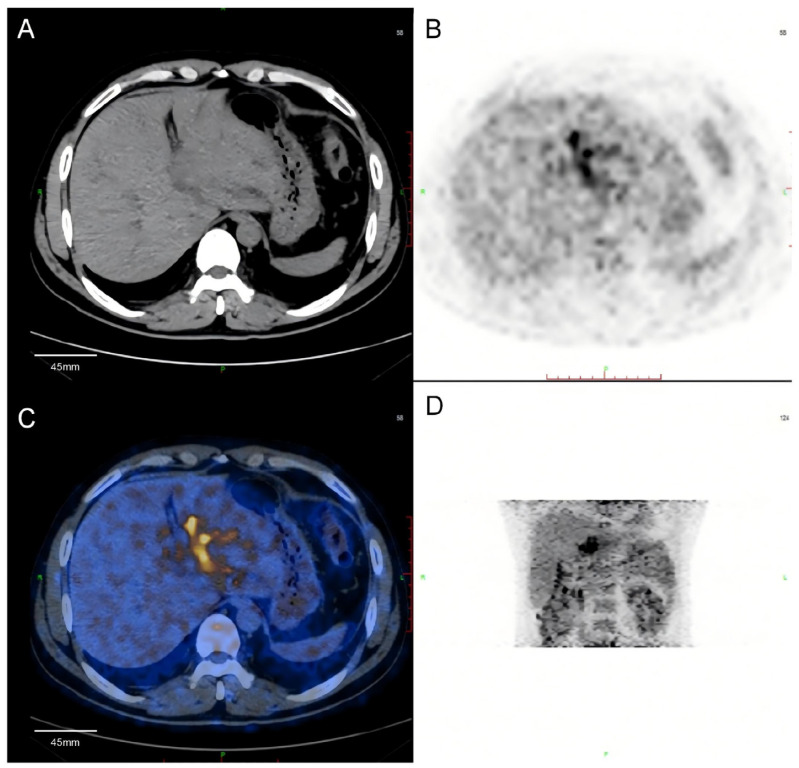
18F-FDG PET-CT images. (**A**) CT shows a low-density mass in the left liver lobe. (**B**) PET shows intense FDG uptake in the lesion (SUVmax 10.8). (**C**) Fused PET-CT confirms the hypermetabolic focus at the hepatic hilum. (**D**) Maximum intensity projection (MIP) shows no abnormal FDG uptake elsewhere in the body. (The color overlay in panel (**C**) (yellow/orange) represents areas of high FDG uptake indicating hypermetabolic regions, overlaid on the blue CT anatomical background. This color-coded fusion imaging is standard in PET-CT to highlight metabolically active tumor regions).

**Figure 4 diagnostics-15-03057-f004:**
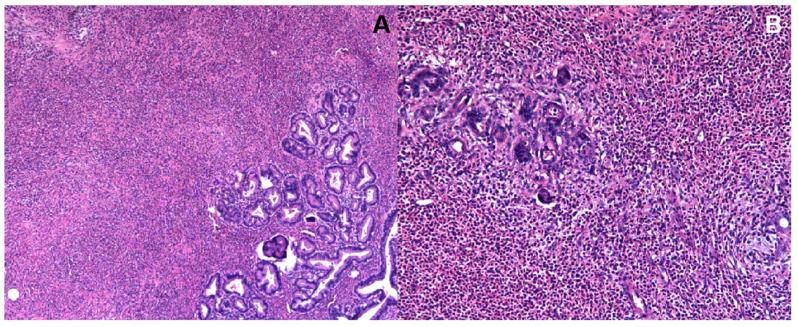
Histopathological examination of the hepatic lesion (H&E staining). (**A**) Low magnification (40×) shows dense infiltration of eosinophils and histiocytes with areas of necrosis. (**B**) High magnification (100×) reveals numerous eosinophils and scattered glandular structures; no evidence of malignancy.

**Table 1 diagnostics-15-03057-t001:** Laboratory findings at admission.

Parameter	Result	Reference Range
Hematology		
White blood cell count	8.5 × 10^9^/L	4.0–10.0 × 10^9^/L
Eosinophil percentage	34%	0.4–8.0%
Hemoglobin	135 g/L	130–175 g/L
Platelet count	245 × 10^9^/L	100–300 × 10^9^/L
Liver Function Tests		
Total bilirubin	51.3 μmol/L	5.1–22.2 μmol/L
Direct bilirubin	28.8 μmol/L	0–6.8 μmol/L
ALT	119 U/L	7–40 U/L
AST	47 U/L	13–35 U/L
ALP	221 U/L	45–125 U/L
GGT	480 U/L	10–60 U/L
Tumor Markers		
CA125	199.7 U/mL	0–35 U/mL
CA19-9	19.8 U/mL	0–37 U/mL
CEA	0.9 ng/mL	0–5 ng/mL
AFP	3.5 ng/mL	0–20 ng/mL
Parasitic Serology	Negative	
Stool ova examination	Negative	

**Table 2 diagnostics-15-03057-t002:** Summary of Reported Cases of Eosinophilic Liver Lesions.

Author, Year	Country	Age/Sex	Eosinophil %	Liver Function	PET-CT (SUVmax)	Initial Diagnosis	Etiology	Treatment	Outcome
Le et al., 2025 [5]	Vietnam	40/F	35.3%	Normal	No	ICC	*Fasciola hepatica*	Triclabendazole	Complete resolution (6 mo)
Sachdeva et al., 2025 [6]	India	42/F	25%	Normal	No	Parasitic	*Toxocara*	Albendazole	80% resolution (6 wk)
Li et al., 2025 [7]	China	48/M	Markedly increased	NR	Yes (FDG & FAPI)	Malignancy	NR	NR	Confirmed ELI
Shigematsu et al., 2018 [8]	Japan	59/M	8.1%	Normal	No	Metastasis	Colon cancer	Colectomy	HEA disappeared (6 mo)
Our case	China	39/M	34%	Cholestasis	Yes (10.8)	ICC	Idiopathic	Hepatectomy	No recurrence (1 yr)

Abbreviations: ELI, eosinophilic liver infiltration; HEA, hepatic eosinophilic abscess; ICC, intrahepatic cholangiocarcinoma; NR, not reported; mo, months; wk, weeks; yr, year.

## Data Availability

The data presented in this study are available on request from the corresponding author.

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
