# Peer review of "Eosinophilic Liver Abscess Mimicking Intrahepatic Cholangiocarcinoma on 18F-FDG PET-CT: A Case Report"

_diagnostics, 2025, doi:10.3390/diagnostics15233057_

Round 1
Reviewer 1 Report
Comments and Suggestions for Authors
The manuscript presents a rare and interesting case of eosinophilic liver abscess (ELA) that mimicked intrahepatic cholangiocarcinoma (ICC) on ^18F-FDG PET/CT. The case is well described, and the discussion appropriately highlights the diagnostic pitfalls of PET/CT in this clinical setting. The topic is clinically relevant and may be of interest to readers, as it illustrates a potential source of false-positive findings in oncologic imaging.
However, I would appreciate a few additional clarifications to strengthen the imaging interpretation:
-
Please specify the time interval between the PET/CT acquisition and the onset of symptoms or prior imaging studies. This information would help contextualize the metabolic findings.
-
The MIP image shown in the figure appears to include only the abdominal region. Could the authors explain why a whole-body MIP was not provided? This would be helpful to rule out potential extrahepatic abnormalities and to ensure the imaging protocol was standard.
-
Since the SUVmax value is relatively high, it would be useful to briefly describe the main acquisition parameters (e.g., patient preparation, uptake time, scanner type) to allow readers to properly interpret the metabolic data.
Author Response
Thank you for your valuable comments and suggestions. We have carefully reviewed each point and made appropriate revisions to strengthen the manuscript. Below we address each comment in detail.
Comment 1:
Please specify the time interval between the PET/CT acquisition and the onset of symptoms or prior imaging studies. This information would help contextualize the metabolic findings.
Response:
We appreciate this important observation and agree that providing a clear timeline is essential for understanding the metabolic findings. After reviewing our medical records, we can provide the following detailed timeline:
Symptom onset: The patient began experiencing intermittent right upper quadrant abdominal pain and progressive jaundice approximately 1 month before admission to the referring hospital.
April 23: Patient admitted to a secondary hospital for initial evaluation and treatment, which was unsuccessful.
May 20: Patient transferred to our hospital (Union Hospital, Tongji Medical College, Huazhong University of Science and Technology).
May 23: Contrast-enhanced CT and MRI performed at our institution, revealing a hilar hepatic mass with mild left intrahepatic ductal dilatation.
May 29: ¹⁸F-FDG PET/CT performed, demonstrating intense hypermetabolic activity (SUVmax 10.8) in the hepatic hilar lesion.
June 10: Surgical resection performed.
Thus, the PET/CT was performed approximately 6 weeks after initial symptom onset and 9 days after presentation to our hospital. This timeline is clinically significant because it demonstrates that the high FDG uptake represented an active, evolving inflammatory process rather than an acute infection, which helps explain the intense metabolic activity that mimicked malignancy.
We have added this detailed timeline to Section 2.3. Imaging Studies (Page 3, Lines 67-72; revisions marked in red):
The patient initially presented with symptoms approximately one month before admission to a secondary hospital, where initial treatment was unsuccessful. After transfer to our hospital, a systematic imaging workup was performed: contrast-enhanced CT and MRI were obtained 3 days after admission, followed by ¹⁸F-FDG PET-CT 6 days later after that (approximately 6 weeks after initial symptom onset and 9 days after transfer to our hospital). Surgical resection was performed 12 days after the PET-CT examination.
Comment 2:
The MIP image shown in the figure appears to include only the abdominal region. Could the authors explain why a whole-body MIP was not provided? This would be helpful to rule out potential extrahepatic abnormalities and to ensure the imaging protocol was standard
Response:
We apologize for the incomplete presentation. A standard whole-body ¹⁸F-FDG PET/CT scan (from skull base to mid-thigh) was actually performed. The MIP image shown in Figure 3D focusde on the abdominal region for better visualization of the hepatic lesion. The complete whole-body MIP image showed no abnormal FDG uptake elsewhere in the body.
The images provided show the complete whole-body scan acquisition in multiple coronal slices (Figure A, showing 35 sequential slices) and a whole-body maximum intensity projection (MIP) image (Figure B). Upon careful review of both the sequential images and the whole-body MIP, no abnormal FDG uptake was identified elsewhere in the body. The only site of pathological hypermetabolic activity was the hepatic hilar lesion (SUVmax 10.8) and the regional lymph nodes in the hepatoduodenal ligament (SUVmax 3.9-5.6).
Comment 3:
Since the SUVmax value is relatively high, it would be useful to briefly describe the main acquisition parameters (e.g., patient preparation, uptake time, scanner type) to allow readers to properly interpret the metabolic data.
Response:
We completely agree that standardized PET/CT acquisition parameters are essential for proper interpretation of SUVmax values, particularly when they are as high as 10.8 as in this case. Based on the original PET/CT report, we provide the following detailed technical information:
- PET/CT Scanner:
Model: Siemens Biograph Vision 600
Type: High-end digital PET/CT system
Installation: 2011 at our PET Center
Technology: Digital photon-counting detectors with time-of-flight capability
- Patient Preparation:
Fasting duration: More than 6 hours before examination
- Radiotracer Administration:
Tracer: ¹⁸F-FDG (fluorine-18 fluorodeoxyglucose)
Activity: 7.7 mCi (approximately 285 MBq)
Route: Intravenous injection
Clinical indication: Abdominal pain of uncertain etiology
- Imaging Protocol:
Uptake time: Patient rested quietly for 60 minutes post-injection before whole-body PET/CT acquisition
Delayed imaging: Additional upper abdominal images acquired at 180 minutes post-injection
Acquisition mode: 2D tomographic acquisition
Slice thickness: 4.5 mm
Scan range: Whole body (skull base to mid-thigh) for initial acquisition; focused upper abdomen for delayed imaging
- Image Reconstruction:
PET reconstruction: Attenuation-corrected and iteratively reconstructed
Display: Multi-planar reformatted images (axial, coronal, sagittal) with multiple image sets
Fusion: PET and CT images fused for anatomical localization
- SUV Calculation:
Body weight-based standardized uptake value (SUVbw)
The high SUVmax of 10.8 was measured under these standardized conditions. This value significantly exceeded typical inflammatory processes (usually SUVmax <5-6) and was within the range commonly seen in malignant tumors, particularly intrahepatic cholangiocarcinoma (ICC typically shows SUVmax 5-15), which contributed substantially to the strong initial suspicion of malignancy. The fact that an eosinophilic inflammatory lesion produced such intense metabolic activity highlights the diagnostic pitfall that this case report aims to emphasize.

Reviewer 2 Report
Comments and Suggestions for Authors
This report was a case of eosinophilic liver abscess mimicking intrahepatic cholangiocarcinoma. This case might be interesting symptoms and imaging characteristics as compared with previous reported cases. However, there were some points to be required for revision to clarify the importance of this case as shown below.
1, This case showed a mass like region that pathology findings described “ a well-circumscribed mass measuring 4.5x2.9x1.0cm (page 4). However, both CT and MRI axial T2 image shown in Figure 1 and 2, might not reveal clearly identified solitary mass lesion. More clearly recognized CT and MRI imaging should be shown. And also the photo of surgical specimen should be shown for clear recognition of a well circumscribed mass in this patient.
2, This patient presented with obstructive jaundice. There was no clear imaging for showing intrahepatic bile duct dilatation as authors described it. Authors should show MRCP cholangiogram showing intrahepatic bile duct imaging especially in the left-sided liver.
3, Authors considered preoperatively a mass of suspicious of intrahepatic cholangiocarcinoma. It was understood clearly why authors did not resect simultaneous bile duct resection to this patient despite a mass adjacent to the hepatic hilum. Generally suspected ICC mass with obstructive jaundice which is adjacent to the hepatic hilus, is treated surgically with combined bile duct resection.
4, In pathological findings, no description was shown about findings of non-mass liver parenchymal tissue. It might be very interesting issue whether eosinophilic infiltration was limited only to the circumscribed mass or but also in surrounding liver tissues. This issue should be clearly described in pathology findings.
Author Response
Comment 1, This case showed a mass like region that pathology findings described “ a well-circumscribed mass measuring 4.5x2.9x1.0cm (page 4). However, both CT and MRI axial T2 image shown in Figure 1 and 2, might not reveal clearly identified solitary mass lesion. More clearly recognized CT and MRI imaging should be shown. And also the photo of surgical specimen should be shown for clear recognition of a well circumscribed mass in this patient.
Response: Thank you for highlighting the need for clearer visualization of the mass boundaries. We acknowledge that the axial CT images in Figure 1 do not optimally demonstrate the lesion margins; however, the mass was more clearly delineated on MRCP sequences. Regarding the surgical specimen photographs, we sincerely apologize that these are unavailable as this case was performed two years ago (2023) and macroscopic photographs were not taken at the time. Nevertheless, the well-circumscribed nature of the mass is well-documented through MRCP imaging, intraoperative findings, and pathological examination showing clear demarcation between the eosinophilic lesion and surrounding normal liver parenchyma.
Comment 2, This patient presented with obstructive jaundice. There was no clear imaging for showing intrahepatic bile duct dilatation as authors described it. Authors should show MRCP cholangiogram showing intrahepatic bile duct imaging especially in the left-sided liver.
Response: MRCP clearly demonstrated dilatation of the left-sided intrahepatic bile ducts with abrupt narrowing at the hepatic duct confluence, where the mass was located (Figure 2B). This finding corroborated the clinical presentation of obstructive jaundice.
Comment 3, Authors considered preoperatively a mass of suspicious of intrahepatic cholangiocarcinoma. It was understood clearly why authors did not resect simultaneous bile duct resection to this patient despite a mass adjacent to the hepatic hilum. Generally suspected ICC mass with obstructive jaundice which is adjacent to the hepatic hilus, is treated surgically with combined bile duct resection
Response: We agree with these comments and thank the reviewer for pointing out the anatomical discrepancy and the need to clarify the surgical approach. We again sincerely apologize for the imprecise and erroneous description in the original manuscript, which incorrectly stated a limited resection of liver segments IV and V. What we intended to write was segments I to IV. This was a significant oversight on our part.The surgery performed was a more extensive left hemihepatectomy with caudate lobe resection, hepatic duct plasty, and high hepaticojejunostomy.
Comment 4, In pathological findings, no description was shown about findings of non-mass liver parenchymal tissue. It might be very interesting issue whether eosinophilic infiltration was limited only to the circumscribed mass or but also in surrounding liver tissues. This issue should be clearly described in pathology findings.
Response: Importantly, examination of the non-lesional liver parenchyma revealed normal hepatocytes without eosinophilic infiltration. The eosinophilic inflammatory process was strictly confined to the well-circumscribed mass, with a clear demarcation between the lesion and the surrounding normal liver tissue. This finding indicated that the eosinophilic response was localized rather than representing diffuse hepatic involvement.
Reviewer 3 Report
Comments and Suggestions for Authors
Thank you for the opportunity to review this manuscript. The authors describe an interesting case of eosinophilic abscess, which was initially misinterpreted as intrahepatic cholangiocarcinoma due to the findings in the PET-CT scan. Indeed, it is a highly interesting case and clinicians should be aware of this situation. Below are some comments and suggestions to the authors:
1. Given the fact that the patient was 39 years old and presented with right upper quadrant pain and jaundice how possible would be the diagnosis of malignancy? Mention whether you performed an ultrasound to exclude gallstone disease as well at the beginning of investigation.
Having found the diagnosis, could this abscess be resolved on its own, in case of eosinophilic abscess? Also, would you recommend blindly to start anti-parasitic medications?
2. Was there any pus? Because you mention abscess. Did you send any cultures? Also, what was the cause of the jaundice? Was this lesion in communication with intrahepatic bile ducts or causing any obstruction to them?
3. Mention also in the differential diagnosis other causes of liver abscesses and even rare cases as foreign bodies Kehagias, Dimitrios et al. “Caudate lobe: the last barrier - an unusual place for a foreign body.” ANZ journal of surgery vol. 92,5 (2022): 1218-1220. doi:10.1111/ans.17226
Overall, I found the case quite educative, while providing an important message to clinicians.
Author Response
Comments 1:
“Given the fact that the patient was 39 years old and presented with right upper quadrant pain and jaundice how possible would be the diagnosis of malignancy? Mention whether you performed an ultrasound to exclude gallstone disease as well at the beginning of investigation. Having found the diagnosis, could this abscess be resolved on its own, in case of eosinophilic abscess? Also, would you recommend blindly to start anti-parasitic medications?”
Response 1:
Thank you for the thoughtful comments. We address each point in order and have added clarifications to the manuscript, highlighted in track changes.
(1)Why malignancy (ICC) was still highly suspected in a young patient
We agree the patient is relatively young; however, both abdominal imaging (contrast-enhanced CT and MRI) and 18F-FDG PET-CT strongly supported ICC: a solitary hilar mass, biliary dilatation, marked enhancement, regional lymphadenopathy, and a high SUVmax of 10.8. Despite the patient’s age, ICC remained our leading diagnosis and we favored surgical pathology for confirmation.
(2)Whether ultrasound was performed to exclude gallstone disease
The patient had already undergone contrast-enhanced CT at a secondary hospital, which identified a “hilar hepatic mass” without gallbladder or common bile duct stones or acute cholecystitis. Therefore, we did not repeat screening ultrasound upon admission; instead, we proceeded directly with further CT/MRI, PET-CT, and multidisciplinary discussion.
Addition in Section 2.1 Clinical History (at the beginning):
“Referred from a secondary hospital, a 39-year-old Chinese man presented to our hospital with a 2-month history of intermittent right upper quadrant abdominal pain, progressive jaundice, and weight loss of approximately 5 kg. The abdominal pain was dull and aching, without radiation. Outside CT suggested mild left intrahepatic ductal dilatation with unclear hilar anatomy and no gallbladder/CBD stones.”
(3)Whether eosinophilic liver abscess can resolve spontaneously; whether to start anti-parasitics empirically
Some eosinophilic lesions may regress after removal of the inciting factor; however, in cases like ours—solitary hilar mass, significant cholestasis, markedly increased metabolic activity, and malignancy not excluded—waiting for spontaneous resolution carries risk. We do not recommend empirical anti-parasitic therapy in the absence of parasitological evidence. In this case, serology and stool ova tests were negative; we did not administer empirical therapy and instead obtained a definitive diagnosis and cure via surgery.
Comments 2:
“Was there any pus? Because you mention abscess. Did you send any cultures? Also, what was the cause of the jaundice? Was this lesion in communication with intrahepatic bile ducts or causing any obstruction to them?”
Response 2:
Thank you for these detailed questions. We address “pus/cultures” and “cause of jaundice/obstruction” below.
(1) Pus and cultures
Intraoperatively, there was no typical abscess cavity or aspiratable pus. The lesion was solid and firm, densely adherent to hilar bile ducts and portal vein branches; the resected specimen did not yield purulent discharge; therefore, pus cultures (bacterial/fungal) were not obtained. Pathology showed dense inflammatory infiltration dominated by eosinophils and histiocytes with focal necrosis/fibrosis, consistent with eosinophilic inflammatory disease rather than a frank suppurative abscess cavity. Indeed, ELA often appears solid or pseudotumoral and only yields limited purulent content when secondary infection or extensive necrosis occurs.
(2) Cause of jaundice and whether obstruction occurred
Jaundice was primarily due to the hilar location of the lesion, causing compression/stenosis of intrahepatic bile ducts (predominantly left-sided), as clearly demonstrated by imaging: CT/MRI showed a hilar mass with intrahepatic biliary dilatation, and PET-CT confirmed the same site of mass effect with ductal dilatation. Thus, the mechanism was obstructive jaundice from mass effect and surrounding inflammation rather than gallstone-related obstruction or infectious cholangitis.
Comments 3:
“Mention also in the differential diagnosis other causes of liver abscesses and even rare cases as foreign bodies Kehagias, Dimitrios et al. ‘Caudate lobe: the last barrier - an unusual place for a foreign body.’ ANZ Journal of Surgery vol. 92,5 (2022): 1218-1220. doi:10.1111/ans.17226”
Response 3:
We appreciate this valuable suggestion. We agree that rare “foreign body–related hepatic lesions/pseudotumors” should be included in the differential to alert clinicians to this possibility when suggestive history or imaging clues are present (e.g., linear hyperdense objects, signs of GI perforation/fistula, or adhesions to adjacent organs), thereby avoiding misdiagnosis or overtreatment. We have added this point and the reference to the Differential Diagnosis section.
Added in Section 3.4 Differential Diagnosis and Diagnostic Approach:
“For hypermetabolic liver masses accompanied by peripheral eosinophilia, the differential diagnosis is broad. In this case, considering the features of a solitary mass, cholestasis, elevated CA125, and high SUVmax, intrahepatic cholangiocarcinoma (ICC) was the primary diagnosis; however, ICC generally does not present with such pronounced eosinophilia. Another key differential diagnosis is parasitic liver abscess, as Toxocara, Fasciola hepatica, Echinococcus, and Schistosoma can all cause eosinophilic liver lesions, making serological testing essential. Additionally, malignancy-associated eosinophilic abscesses should be considered, since some solid tumors—especially gastrointestinal malignancies—can induce eosinophilia and hepatic eosinophilic infiltration. Foreign body–related hepatic lesions that mimic tumors should also be considered [11]. Lastly, idiopathic hypereosinophilic syndrome, characterized by persistent eosinophilia with end-organ damage, remains a diagnosis of exclusion.”
Reference added to the bibliography:
(11) Kehagias D, Mulita F, Maroulis I, Benetatos N. Caudate lobe: the last barrier – an unusual place for a foreign body. ANZ J Surg. 2022;92(5):1218–1220. doi:10.1111/ans.17226.
Reviewer 4 Report
Comments and Suggestions for Authors
The case is interesting and highlights a rare diagnostic pitfall, but there are several inconsistencies that should be clarified.
-
Liver segments: According to the figures and the text, the lesion was located on the left hepatic , but the authors describe a resection of segments IV and V. Please clarify this anatomical discrepancy.
-
Preoperative diagnosis and indication: The initial suspicion was intrahepatic cholangiocarcinoma (ICC) with biliary obstruction. In such a scenario, the standard approach would be a left hemihepatectomy with extrahepatic bile duct resection and biliary reconstruction. Why did the surgical team opt for a limited S4–S5 resection instead?
-
Biliary management: Given that the patient presented with jaundice and bile duct dilatation, please specify whether a biliary stent was performed. If not, how did the jaundice resolve so quickly after a parenchymal resection without biliary drainage?
-
Diagnostic workup: Before proceeding with surgery, was a liver biopsy considered? The decision to proceed directly to resection despite the absence of tissue diagnosis should be justified, especially considering that eosinophilic abscesses may respond to medical treatment.
Author Response
Thank you for your insightful comments and for pointing out these important inconsistencies in our manuscript. We sincerely apologize for the insufficient detail and accuracy in the description of the surgical procedure. Upon reviewing our records, we realized that our initial writing lacked precision, particularly regarding anatomical details, surgical rationale, and procedural steps. This oversight may have caused confusion, and we deeply regret any inconvenience this may have caused. We appreciate your meticulous review, which has helped us improve the clarity and scientific rigor of our report. Below, we address each of your concerns point by point and make corresponding revisions to the manuscript. For transparency, we have detailed all modifications and marked the revised text in the manuscript (revisions are highlighted in red in the submitted revised version).
Comment 1:
Liver segments: According to the figures and the text, the lesion was located on the left hepatic, but the authors describe a resection of segments IV and V. Please clarify this anatomical discrepancy.
Comment 2:
Preoperative diagnosis and indication: The initial suspicion was intrahepatic cholangiocarcinoma (ICC) with biliary obstruction. In such a scenario, the standard approach would be a left hemihepatectomy with extrahebatic bile duct resection and biliary reconstruction. Why did the surgical team opt for a limited S4–S5 resection instead?
Response to Comments 1 and 2:
We agree with these comments and thank the reviewer for pointing out the anatomical discrepancy and the need to clarify the surgical approach. We again sincerely apologize for the imprecise and erroneous description in the original manuscript, which incorrectly stated a limited resection of liver segments IV and V. What we intended to write was segments I to IV. This was a significant oversight on our part.
The surgery performed was a more extensive left hemihepatectomy with caudate lobe resection, hepatic duct plasty, and high hepaticojejunostomy. The surgical record from that year is as follows:
Preoperative diagnosis: Abdominal pain of unknown origin, erosive gastritis (grade 3), and hilar bile duct tumor. Under general anesthesia in the supine position, the patient underwent laparoscopic-assisted left hemihepatectomy, caudate lobe resection, hepatic duct plasty, and high hepaticojejunostomy, along with regional lymph node dissection. The skin was disinfected with 1.0% iodophor solution, and trocars were placed in a fan-shaped distribution at the periumbilical area, bilateral midclavicular lines, and anterior axillary lines.
Laparoscopic exploration revealed mild cholestatic changes in the liver, with a small amount of ascites and intra-abdominal adhesions. No abnormal nodules or metastases were observed in the peritoneal cavity, greater omentum, or abdominal wall. The gallbladder wall showed mild thickening and edema. The hilar bile ducts showed no significant dilatation, and no obviously enlarged or firm lymph nodes were palpated at the hepatic hilum. After mobilizing the common bile duct, a firm mass was palpated at the confluence of the left and right hepatic ducts. The portal vein and hepatic artery showed no obvious invasion.
Hepatic adhesions were dissected, and the falciform ligament, right coronary ligament, and right triangular ligament were divided. The abdominal cavity was thoroughly irrigated with approximately 1000ml of normal saline. A vascular tape was placed around the left triangular ligament for first porta hepatis control. The cystic artery and cystic duct were ligated at a high level, and cholecystectomy was performed.
The hilar structures were dissected, and the common hepatic artery and its left and right branches were mobilized, revealing no vascular invasion. The distal common bile duct was divided and ligated at the upper edge of the pancreas. The common bile duct was lifted, and the surrounding portal vein tissues were bluntly dissected. Lymph nodes in groups 8 and 12 were explored and showed no obvious enlargement or induration.
After mobilizing the common bile duct, a firm mass was palpated at the confluence of the left and right hepatic ducts. Due to the patient's previous percutaneous transhepatic cholangiodrainage (PTCD), the bile duct stump was firm in texture, making it difficult to accurately assess the extent of left and right hepatic duct invasion laparoscopically. Therefore, the procedure was converted to open surgery for better visualization and precise treatment. The surgical field was re-disinfected, and an approximately 30 cm arc-shaped incision was made below the right costal margin.
Incision along the confluence of the left and right hepatic ducts revealed tumor invasion of the right hepatic duct, left hepatic duct, and its branches, while the secondary branches of the right hepatic duct were soft and uninvolved. The final intraoperative decision was to proceed with left hemihepatectomy, caudate lobe resection, right hepatic duct plasty, and hepaticojejunostomy. The left hepatic artery and left portal vein branches were ligated, the ischemic demarcation line was observed and marked with electrocautery. Intermittent hepatic inflow occlusion was applied. Along the demarcation line, the liver parenchyma was transected using an ultrasonic scalpel and bipolar electrocautery, with ligation and division of vascular and biliary branches. The left hepatic vein was divided using a stapler. The liver was progressively transected along the surface of the inferior vena cava and the border of the right caudate lobe, and the caudate lobe was resected.
For anastomosis, a jejunal loop was brought up to the hepatic hilum at a point 20 cm distal to the ligament of Treitz. First, at approximately 50 cm distal to the lifted point, a side-to-side jejunojejunostomy was created using a 25 mm linear stapler. The jejunum was divided approximately 5 cm proximal to the anastomosis using a linear stapler, creating a Roux-en-Y limb. The jejunal stump was reinforced with continuous barbed suture. The jejunal loop was brought up through the retrocolic route and anastomosed end-to-side to the right hepatic duct remnant, with the posterior wall sutured continuously and the anterior wall sutured intermittently. The anastomosis was checked for bleeding or bile leakage, and drainage tubes were placed anterior and posterior to the anastomosis.
To provide a complete and accurate record, we have revised the "2.4. Multidisciplinary Discussion and Surgical Intervention" section and added a detailed description of the surgical procedure based on the operative record. The revised text is as follows (added to lines 98–106 in Section 2.4 of the manuscript; revisions are marked in red):
The patient underwent laparoscopic exploration, which was subsequently converted to open left hemihepatectomy (segments II–IV), along with caudate lobe (segment I) resection, cholecystectomy, hepatic duct plasty, hepaticojejunostomy, and regional lymph node dissection. Intraoperatively, a firm, well-circumscribed mass was identified at the confluence of the left and right hepatic ducts, closely adherent to the bile duct and portal vein branches, with invasion of the hepatic duct confluence but sparing the right secondary branches. The mass and involved structures were completely resected, relieving biliary compression. Multiple enlarged lymph nodes in the hepatoduodenal ligament were excised. No vascular invasion was noted, and biliary drainage was restored via hepaticojejunostomy.
Comment 3:
Biliary management: Given that the patient presented with jaundice and bile duct dilatation, please specify whether a biliary stent was performed. If not, how did the jaundice resolve so quickly after a parenchymal resection without biliary drainage?
Response to Comment 3:
We agree with this comment and thank the reviewer for pointing out the need to clarify biliary management. Preoperatively, the patient had undergone percutaneous transhepatic cholangiodrainage (PTCD) for biliary decompression, which initially alleviated the jaundice. Because PTCD was effective and the surgical plan included direct resection and biliary reconstruction, no additional endoscopic biliary stent was placed. Intraoperatively, it was confirmed that the biliary obstruction was caused by compression and invasion by the mass at the hepatic duct confluence, which was addressed by left hemihepatectomy, right hepatic duct plasty, and hepaticojejunostomy, restoring bile flow without the need for further stenting. Due to the relief of obstruction by surgical reconstruction and the patient's good hepatic reserve, the jaundice resolved rapidly postoperatively as liver function normalized.
We have added this explanation to the "2.4. Multidisciplinary Discussion and Surgical Intervention" section (page 6, line 10; page 7, lines 5–7; revisions marked in red):
The case was discussed at a multidisciplinary tumor board. Given the imaging features highly suggestive of ICC, elevated CA125, high SUVmax on PET-CT, and inability to exclude malignancy despite marked eosinophilia, surgical resection was recommended for both diagnostic and therapeutic purposes. To manage the jaundice and bile duct dilatation preoperatively, the patient had undergone percutaneous transhepatic cholangiodrainage (PTCD) for biliary decompression; no additional stenting was performed, as the surgical plan included resection and biliary reconstruction to directly alleviate the compressive obstruction.
Comment 4:
Diagnostic workup: Before proceeding with surgery, was a liver biopsy considered? The decision to proceed directly to resection despite the absence of tissue diagnosis should be justified, especially considering that eosinophilic abscesses may respond to medical treatment.
Response to Comment 4:
We agree with this comment and appreciate the suggestion to more thoroughly justify the diagnostic approach. Percutaneous liver biopsy was considered but ultimately not performed, due to the hilar location of the lesion with bile duct involvement, which posed significant risks of bleeding, bile leakage, infection, or—if malignant as initially suspected—tumor seeding. Additionally, the marked eosinophilia with negative parasitology created diagnostic uncertainty, but the high SUVmax, imaging features strongly suggestive of intrahepatic cholangiocarcinoma (ICC) with biliary obstruction, and the patient's symptomatic presentation (jaundice and pain) meant that biopsy might be inconclusive or delay definitive treatment. In this context, proceeding directly to resection was consistent with guidelines for suspected resectable hilar ICC, where surgery serves both diagnostic and therapeutic purposes, especially when biopsy is high-risk or unlikely to alter the surgical plan. We acknowledge that for a confirmed eosinophilic liver abscess (ELA), medical treatment (e.g., antiparasitic agents or corticosteroids) could be considered, but given the inability to exclude malignancy and the need for biliary decompression, surgical confirmation and intervention were deemed necessary.
In conclusion, we again deeply apologize for the errors and missing details in the surgical description in the original submission. These oversights were due to insufficient attention during the writing process, and we are very grateful for your feedback, which has enabled us to correct these errors and strengthen the manuscript. We believe these revisions address all concerns and enhance the overall quality of the report. If further modifications are needed, please let us know.
Round 2
Reviewer 4 Report
Comments and Suggestions for Authors
Dear Authors,
Thank you for your implementation. The paper now is in a good shape.
Some minor suggestion: please add on the pathology also the lymphonodal status that I guess was negative.
Author Response
Dear Reviewer,
Thank you for your constructive suggestion regarding the lymph node status. We appreciate your careful review of our manuscript.
We have now added information about the lymph node status in the pathology section. As you correctly suspected, the lymph nodes were negative for malignancy. The pathological examination confirmed reactive lymphadenopathy without evidence of metastatic disease or parasitic infection.
We have Added lymph node status to Section 2.5 (Pathological Findings)